# Unusual Ileocecal Ulcers after Liver Transplantation for Hepatitis B Cirrhosis and Hepatocellular Carcinoma

**DOI:** 10.3390/diagnostics12112668

**Published:** 2022-11-02

**Authors:** Kang He, Zhifeng Xi, Qiang Xia

**Affiliations:** 1Department of Liver Surgery, Renji Hospital, School of Medicine, Shanghai Jiao Tong University, Shanghai 200127, China; 2Shanghai Engineering Research Center of Transplantation and Immunology, Shanghai 200127, China; 3Shanghai Institute of Transplantation, Shanghai 200127, China

**Keywords:** liver transplantation, ileocecal ulcer, tacrolimus

## Abstract

We presented a case demonstrating ileocecal ulcers after liver transplantation for hepatitis B cirrhosis and hepatocellular carcinoma. The patient presented 4 years post-transplant with paroxysmal right lower abdominal pain. Due to a mild increase in the leukocyte and neutrophil count, infectious diseases were initially suspected. However, probiotic treatment did not help improve the symptom. An enhanced CT scan revealed a thickening in the ileocecal region, and endoscopy later showed multiple giant and deep ulcers in the ileocecal region with polypoid hyperplasia. Histopathology of an ulcer biopsy displayed benign ulcers, and chronic inflammation with non-caseous granulomas, without signs of a fungus or parasite infection. Epithelial exfoliation with atypical hyperplasia was observed, and a tacrolimus-induced ileocecal ulcer was considered by a pathologist. Clinical manifestation, lab findings, radiology, and pathology characteristics of ulcers were not consistent with the pathogenesis of ischemia, tuberculosis, CMV, EBV, tumor, or inflammatory bowel diseases. Abdominal pain was gradually relieved and subsided with the discontinuation of tacrolimus and corticosteroid administration.

A 39-year-old female with a history of HBV cirrhosis and hepatocellular carcinoma (within the Milan Criteria) received an orthotopic liver transplant in 2015. The post-transplant course was complicated by a recurrent biliary stricture treated two years later with ERCP. Neither hepatitis B nor hepatocellular carcinoma recurred after liver transplantation until now. However, the patient presented with paroxysmal right lower abdominal pain 4 years post-transplant. Coagulation function parameters were within the normal range. Laboratory indices on the hepatic, renal, cardiac, and respiratory functions were all normal. Blood and stool cultures for bacteria and fungus displayed no growth. Serological results and nucleic acids of CMV and EBV were negative. In addition, negative findings were revealed in the abdominal Doppler ultrasound. Probiotics and antispasmodic treatment did not help improve the symptom, even though a complete blood count revealed a mild increase in the leukocyte and neutrophil count. An enhanced CT scan revealed a thickening of the ileocecal region with the suspicion of inflammatory lesions (Figure 1A). Endoscopy in September 2019 showed multiple giant and deep ulcers in the ileocecal region with polypoid hyperplasia (Figure 1B). To rule out PTLD, a relatively common post-transplant complication, we performed PET-CT at the same time, ^18^F-FDG accumulation in the site of the wall thickening of the ileocecal region (Figure 1C). At that time, her labs showed well graft functions with tacrolimus and mycophenolate as immunosuppressants. Considering the common adverse gastrointestinal effects of mycophenolate after transplant, we empirically discontinued mycophenolate without any improvement in the abdominal pain.

Histopathology of the ulcer biopsy displayed benign ulcers and chronic inflammation with non-caseous granulomas (Figure 2A,B) without signs of a fungus or parasite infection. Epithelial exfoliation with atypical hyperplasia and apoptotic crypt cells were observed, and a tacrolimus-induced ileocecal ulcer was considered by a pathologist. Clinical manifestation, lab findings, radiology, and pathology characteristics of ulcers were not consistent with the pathogenesis of ischemia, PTLD, tuberculosis, CMV, EBV, HSV, tumor, inflammatory bowel diseases, or GVHD. The abdominal pain was gradually relieved and subsided with the discontinuation of tacrolimus and corticosteroid administration. Cyclosporin A was added due to an increase in liver enzymes, suggesting acute rejection. A colonoscopy and biopsy pathology at year 2 of the follow-up revealed significant healing of the ulcers in the ileocecal region (Figure 2C–F). Neither the recurrence of abdominal pain, acute rejection, nor biliary stricture was observed during the 3 years of follow-up. 

The differentiation of the ileocecal ulcer in post-transplant patients is always challenging [1,2,3,4,5]. Common and uncommon pathogenesis, such as infection, ischemia, inflammatory bowel disease, and post-transplant lymphoproliferative disorders should be considered [6]. Endoscopic and pathologic features of tacrolimus-induced ileocecal ulcers are non-specific [7]. Although certain clinical and histological clues may arouse suspicion of it, as in this case, these will not always be present. Histology in isolation often does not provide a definitive diagnosis of tacrolimus-induced ileocecal ulcers. Hence, a multidisciplinary approach is necessary, as well as the collection of clinical information and drug history [8]. Tacrolimus, as well as other immunosuppressants, such as sirolimus or mycophenolate, can result in digestive tract mucosa injury and the development of ulcers [9,10]. However, the histopathologic changes induced by these immunosuppressants and how to differentially diagnose them have not been well characterized.

Erika Hissong reported that most patients with tacrolimus-related digestive tract injuries have endoscopically evident colitis with a variety of histologic features. Most biopsies show an active colitis pattern with apoptotic crypt cells. However, 40% of cases display chronic active colitis that simulates the characteristics of idiopathic inflammatory bowel disease, which causes lots of trouble with differentiation diagnosis [10]. Moreover, for pathologists, the interpretation of biopsy findings in this setting can be challenging because many patients taking tacrolimus are also administered with other immunosuppressants that can cause intestine or colon ulcers. Clinicians should be aware of this potential diagnosis because ulcers and abdominal pain are often resolved with drug cessation or by decreasing the drug dosage rather than therapy aimed at improving inflammatory bowel disease.

In conclusion, solid organ transplantation recipients are at risk of digestive tract injury, ulcers included. Particularly, severe complications such as bleeding or perforation could negatively impact the long-term outcome and sometimes become fatal. Immunosuppressants, such as tacrolimus, sirolimus, or mycophenolate, should be enrolled in the differential diagnosis. Our case can provide support for the idea that tacrolimus can cause ileocecum ulcers with characteristic patterns of inflammation. Discontinuing it or switching to other immunosuppressants could be the solution to dealing with these conditions. 

## Figures and Tables

**Figure 1 diagnostics-12-02668-f001:**
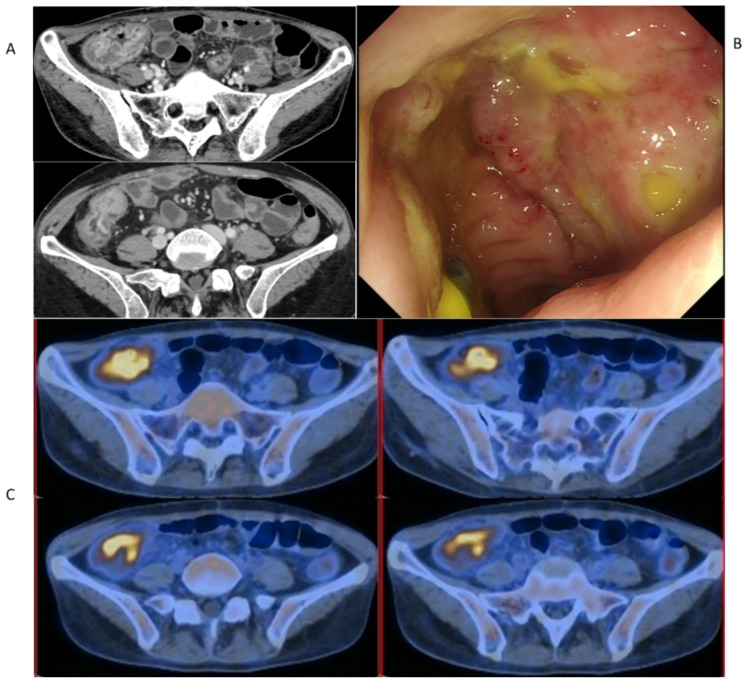
The pre-treatment evaluation of patient’s ileocecal ulcer after admission. (**A**) Enhanced CT scan. (**B**) Endoscopy. (**C**) PET-CT.

**Figure 2 diagnostics-12-02668-f002:**
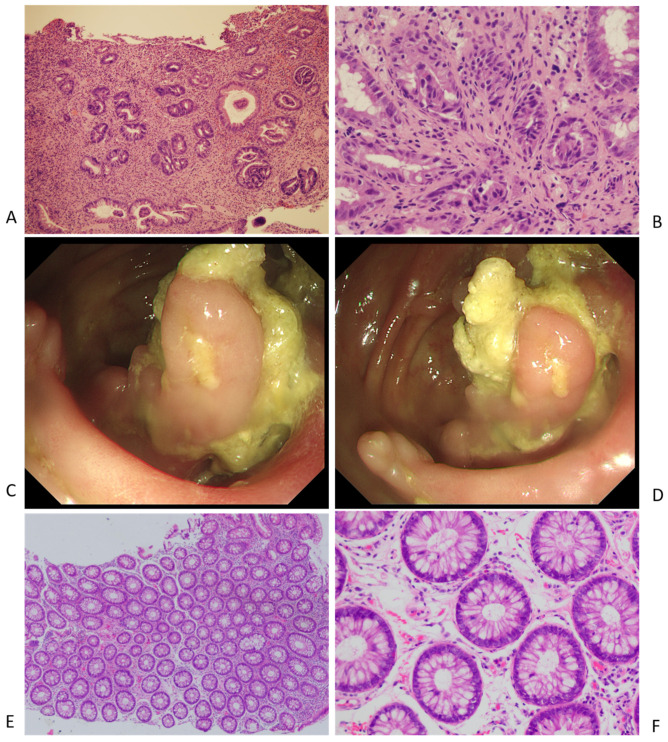
(**A**,**B**) The pre-treatment pathology results of ileocecal ulcer. (**C**,**D**) The post-treatment endoscopy images of ileocecal region. (**E**,**F**) The post-treatment pathology results of ileocecal region.

## Data Availability

Data available on request due to restrictions e.g., privacy or ethical.

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
