# Peer review of "Unusual Ileocecal Ulcers after Liver Transplantation for Hepatitis B Cirrhosis and Hepatocellular Carcinoma"

_diagnostics, 2022, doi:10.3390/diagnostics12112668_

Round 1

Reviewer 1 Report

case report has been written well, it conveys new message that ileocaecal ulcers can occur in post liver transplant patients on tacrolimus, this has been also reported earlier

Author Response

Case report has been written well, it conveys new message that ileocaecal ulcers can occur in post liver transplant patients on tacrolimus, this has been also reported earlier.

Reply: Thank you for your kind advice. However, there are few cases of intestinal ulceration caused by tacrolimus after solid organ transplantation, and only one case has been reported in the literature (World J Gastroenterol. 2016;22(24):5616-22), in which pathological differential diagnosis between tacrolimus and mycophenolate was not done. Instead, they empirically stopped tacrolimus and patient’s ulcer improved. We have revised our manuscript as requested in our discussion. 

Reviewer 2 Report

The authors describe an unusual and interesting side effect of Tacrolimus.  

Are the histopathological findings of the ileocecal ulcers sufficiently distinct to make a definitive diagnosis?

A summary table of prior important publications on intestinal ulcers caused by Tacrolimus in solid organ recipients would add value to the discussion

Mycophenolate induced ulcers in the gastrointestinal tract are more common. This patient was on this drug too. How can one distinguish between the two?

HSV infection, GVHD are other rare possibilities. Were these differentials considered? 

Author Response

Reviewer 2: Comments and Suggestions for Authors

The authors describe an unusual and interesting side effect of Tacrolimus.  

A summary table of prior important publications on intestinal ulcers caused by Tacrolimus in solid organ recipients would add value to the discussion.

Reply: Thank you for your kind advice. However, there are few cases of intestinal ulceration caused by tacrolimus after solid organ transplantation, and only one case has been reported in the literature (World J Gastroenterol. 2016;22(24):5616-22), in which pathological differential diagnosis between tacrolimus and mycophenolate was not done. Instead, they empirically stopped tacrolimus and patient’s ulcer improved. We have revised our manuscript as requested in our discussion.

Are the histopathological findings of the ileocecal ulcers sufficiently distinct to make a definitive diagnosis? Mycophenolate induced ulcers in the gastrointestinal tract are more common. This patient was on this drug too. How can one distinguish between the two?

Reply: Thank you for your advice. Recognition of the histopathologic changes attributable to immunosuppressants has important therapeutic implications, as clinical symptoms are unlikely to resolve unless the offending drug is eliminated. However, these changes have not been well characterized even for commonly used drugs such as tacrolimus or mycophenolate. In our case, when the patient initially presented with abdominal pain, we empirically discontinued mycophenolate, which did not improve her symptoms. Therefore, our diagnosis was more of an exclusion diagnosis and resulted from the significant improvement in symptoms and pathology after discontinuation of tacrolimus. To be noticed is that, in our case, histopathology of ulcer biopsy displayed benign ulcers and chronic inflammation with non-caseous granulomas (Figure 2A and 2B). Epithelial exfoliation with atypical hyperplasia and apoptotic crypt cells were observed and tacrolimus-induced ileocecal ulcer was considered by pathologist.

HSV infection, GVHD are other rare possibilities. Were these differentials considered? 

Reply: Thank you for your advice. We have considered these potential diagnoses, including HSV infection and GVHD. Serologic results and histopathology did not support these causes. We have added this part in our revised manuscript.

Round 2

Reviewer 2 Report

Thank you for addressing all comments. 

The message to the readers can perhaps be that despite the rarity of its presentation, Tacrolimus induced ulcers of the intestine can be an important differential diagnosis in transplant recipients, with therapeutic implications